# Frailty independently predicts unfavorable discharge in non-operative traumatic brain injury: A retrospective single-institution cohort study

**Rahul A. Sastry**[1]*, **Josh R. Feler**[1], **Belinda Shao**[1], **Rohaid Ali**[1], **Lynn McNicoll**[2], **Albert E. Telfeian**[1], **Adetokunbo A. Oyelese**[1], **Robert J. Weil**[3], **Ziya L. Gokaslan**[1]

1 Department of Neurosurgery, Warren Alpert School of Medicine, Rhode Island Hospital, Brown University, Providence, RI, United States of America, 2 Division of Geriatrics, Department of Medicine, Warren Alpert School of Medicine, Brown University, Providence, RI, United States of America, 3 Department of Neurosurgery, Brain & Spine, Southcoast Health, Dartmouth, MA, United States of America

* rahul.sastry@gmail.com

## Abstract

### Background

Frailty is associated with adverse outcomes in traumatically injured geriatric patients but has not been well-studied in geriatric Traumatic Brain Injury (TBI).

### Objective

To assess relationships between frailty and outcomes after TBI

### Methods

The records of all patients aged 70 or older admitted from home to the neurosurgical service of a single institution for non-operative TBI between January 2020 and July 2021 were retrospectively reviewed. The primary outcome was adverse discharge disposition (either in-hospital expiration or discharge to skilled nursing facility (SNF), hospice, or home with hospice). Secondary outcomes included major inpatient complication, 30-day readmission, and length of stay.

### Results

100 patients were included, 90% of whom presented with Glasgow Coma Score (GCS) 14–15. The mean length of stay was 3.78 days. 7% had an in-hospital complication, and 44% had an unfavorable discharge destination. 49% of patients attended follow-up within 3 months. The rate of readmission within 30 days was 13%. Patients were characterized as low frailty (FRAIL score 0–1, n = 35, 35%) or high frailty (FRAIL score 2–5, n = 65, 65%). In multivariate analysis controlling for age and other factors, frailty category (aOR 2.63, 95CI [1.02, 7.14], p = 0.005) was significantly associated with unfavorable discharge. Frailty was

**Data Availability Statement:** Data cannot be shared publicly because they contain confidential

patient information. Data are available from the Rhode Island Hospital Institutional Data Access / Ethics Committee (contact via owen_leary@brown.edu) for researchers who meet the criteria for access to confidential data.

**Funding:** The author(s) received no specific funding for this work.

**Competing interests:** NO The authors have declared that no competing interests exist.

not associated with increased readmission rate, LOS, or rate of complications on uncontrolled univariate analyses.

## Conclusion

Frailty is associated with increased odds of unfavorable discharge disposition for geriatric patients admitted with TBI.

## Introduction

The prevalence of traumatic brain injury (TBI) among elderly patients is significant and likely to increase with the aging of the general population [1–3]. As compared to their younger counterparts, elderly TBI patients have high incidence of comorbid medical diagnoses, increased post-injury morbidity/mortality, slower recovery of pre-injury functional status, and higher likelihood of subsequent re-injury [4–11]. Frailty, defined as the progressive and cumulative decline in physiologic reserve and subsequent increased vulnerability to stressors [12], has been shown to be a reliable marker of in-hospital complications, readmission after trauma, and adverse outcomes in the general surgical literature [13]. For frail geriatric trauma patients, the initiation of integrated care provision plans, which include but are not limited to inpatient geriatric medicine consultation and specialized nursing protocols, have resulted in diminished inpatient delirium, length of stay (LOS), rate of re-admission, and 6-month morbidity and mortality [14–18]. As compared to general multi-system trauma, there is a comparatively small but growing literature base on the topic of frailty in the context of TBI [19–23]. Furthermore, it is unclear if the associations noted between the frailty syndrome and adverse in-hospital and short-term outcomes after general, multi-system trauma are equally applicable to patients with TBI, who may have proportionally greater neurocognitive rather than physiologic deficits after injury. We attempted to evaluate the relationship between frailty and short-term outcomes after non-operative TBI among geriatric patients admitted to a neurosurgical service at a Level I Trauma Center. We hypothesize that frail geriatric patients admitted with TBI are at increased risk of unfavorable discharge outcome, major inpatient complications, increased length of stay, and readmission after discharge from the hospital.

## Methods

The protocol for this study was reviewed and approved by the Institutional Review Board of Rhode Island Hospital (Providence, RI, USA). As the proposed research was a retrospective observational study, the need for patient consent was waived by the aforementioned Institutional Review Board. Data were not fully anonymized at the time of chart review. The methods utilized in this study are in concordance with The Strengthening the Reporting of Observational Studies in Epidemiology (STROBE) guidelines. The records of all patients admitted to the neurosurgery service from the emergency department with an acute traumatic non-operative head trauma between January 2020 and July 2021 were retrospectively identified. At our institution, patients with isolated traumatic brain injury, regardless of operative indication or severity, are admitted to the neurosurgery service when admission is indicated. These patients are drawn broadly from a catchment area that includes Rhode Island, southern Massachusetts, and eastern Connecticut. Patients over the age of 70 are commonly, but not always, administered a questionnaire assessing the FRAIL score, a simple, well-validated, and easily-implemented assessment of frailty that has been used in the orthopedic and general surgical

literature, by neurosurgical residents or advanced practice practitioners [24]. The FRAIL score is a 5-point scale that assesses patient activity tolerance, fatigue with ambulation, fatigue with exertion, number of comorbidities, and recent weight loss equally, 1-point to each. Patients in this study were characterized as being either low frailty (FRAIL score 0–1) or high frailty (FRAIL score 2–5) based on the results of the questionnaire. In situations in which the patient is unable to answer questions, providers typically speak to family members and/or emergency contacts.

Patients admitted from locations other than home and for reasons other than trauma or those admitted after trauma but found to have non-traumatic (i.e. new brain tumor) findings were excluded. As previously noted, patients with operative head trauma were excluded in order to isolate the relative contributions of pre-admission physiologic reserve toward outcomes of interest independently from peri-operative neurologic deficits. Using these criteria, the records of 100 patients were identified. This patient selection process is summarized in **Fig 1**.

All records were evaluated a minimum of three months after hospital admission. Data were collected from the electronic medical record (EMR) and included age, dates of admission and discharge, admission and discharge location and disposition (home, nursing facility, rehabilitation facility), admission level of care (LOC) (intensive care unit (ICU) or floor), ethnicity, gender, FRAIL score, characteristics of injury or injuries present on admission, presence of anticoagulant or antiplatelet medication, number of inpatient consulting services (such as internal medicine, cardiology, general surgery, etc.) consulted, inpatient complications not present on admission (as defined by the National Trauma Data Standard [25]), admission and

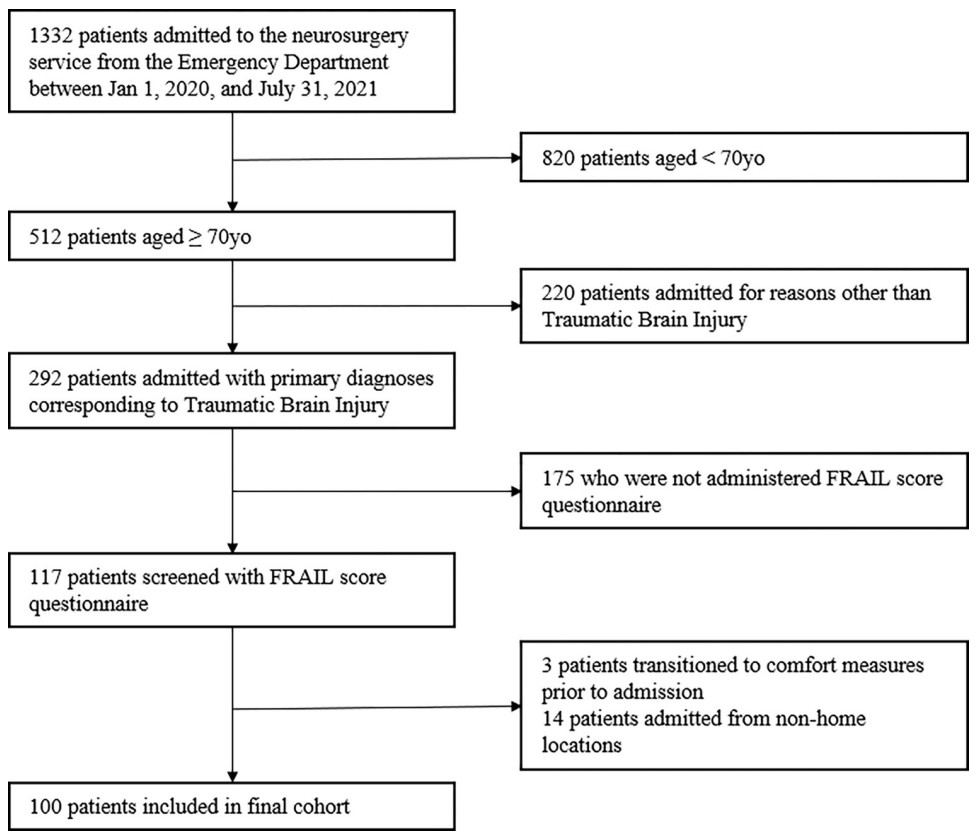

**Fig 1. Patient selection diagram.**

discharge Glasgow Coma Score (GCS), placement of tracheostomy or gastrostomy during admission, re-admission or re-presentation to the emergency department within 30 days of discharge, and follow-up with neurosurgery (either remotely or in-person) within 3 months of discharge. All charts were reviewed by two authors (RAS, JRF) to ensure uniformity of data collection.

The primary outcome of this study was unfavorable discharge disposition, which was defined as discharge to skilled nursing facility (SNF), long-term acute care (LTAC), hospice, home with hospice. Secondary outcomes included rate of major inpatient complications, LOS, and readmission within 30 days. Numerical variables were summarized with mean and standard deviation and categorical variables were reported by proportion excluding missing values. Patients who expired in the hospital were excluded from the denominator of non-applicable outcome variables including length of stay, clinic follow up, discharge GCS, and re-presentation to care. Patients discharged to hospice were excluded from non-applicable outcome variables including clinic follow up variables including clinic follow up and re-presentation to care. LOS was reported by median and range. Univariate comparisons between groups were made with Kruskal-Wallace test. Candidate variables for multivariate modeling of unfavorable discharge disposition were chosen by the authors on the basis of clinical importance and included age, frailty category (binary), presenting GCS category (treated as a categorical variable corresponding to mild [14, 15], moderate [9–13], and severe [3–8] TBI), and antiplatelet/anticoagulant use. Model parameters were given by adjusted odds ratio (aOR), 95% confidence intervals (95CI), and p value. Collinearity was assessed with variance inflation factors (VIF), and all VIF were near 1, so no statistical adjustments were made.

## Results

The characteristics of our patient population are summarized in **Table 1**. The distribution of FRAIL scores among patients included in the analysis is presented in **Fig 2.** Among 100 total patients, 35 (35%) were categorized as Low Frailty and 65 (65%) were categorized as High Frailty. High frailty patients (83.4 years old) were significantly older than low frailty patients (80.1 years old). No significant differences among the three groups were noted with regard to patient sex. Seventy-nine (79%) patients were on antiplatelet medications or anticoagulants at time of admission, and the proportion of patients on these medications was not different between groups (p = 0.173).

Imaging characteristics of TBI are also summarized in **Table 1.** Mixed density subdural hematoma (SDH) was notably more common in high frailty patients (10.8%) as compared to low frailty patients (0%). Acute subdural hematoma (aSDH) (63%) and traumatic subarachnoid hemorrhage (tSAH) (33%) were the most common injury morphologies among included patients. GCS at presentation also did not differ significantly between frailty subcategories. Ninety patients (90%) in this cohort presented with GCS 14 or 15, 4 patients (4%) were intubated on arrival to the hospital, and 90 patients (77.6%) were admitted to floor level of care.

Patient outcomes are summarized in **Table 2.** Seven patients (7.0%) experienced major complications during inpatient admission. The rate of complication development did not vary significantly between frailty categories. High frailty patients (10.9%) were significantly more likely than low frailty patients (0%) to be transferred to another hospital service prior to discharge (*p* = 0.042).

There was no significant difference noted between the number of consulting services per patient, the percentage of patients who were either readmitted or who re-presented to the emergency department within 30 days of discharge, or the percentage of patients who presented to neurosurgical follow up within 30 days.

**Table 1. Patient characteristics of patient cohort at time of admission to the hospital.**

|  | Low Frailty (N = 35) | High Frailty (N = 65) | Total (N = 100) | p value |
|---|---|---|---|---|
| **Age (years)** | 80.1 | 83.4 | 82.2 | **0.032** |
| **Male Gender** | 15 (42.9%) | 28 (43.1%) | 43 (43.0%) | **0.983** |
| **Concomitant Spine Trauma** | 6 (17.1%) | 5 (7.7%) | 11 (11.0%) | **0.150** |
| **Use of either antiplatelet or anticoagulant at time of evaluation** | 25 (71.4%) | 54 (83.1%) | 79 (79.0%) | **0.173** |
| **Anticoagulant** | 10 (28.6%) | 20 (30.8%) | 30 (30.0%) | **0.819** |
| **Antiplatelet** | 18 (51.4%) | 42 (64.6%) | 60 (60.0%) | **0.199** |
| **CT[1] Findings** |  |  |  |  |
| **Acute SDH[2]** | 23 (65.7%) | 40 (61.5%) | 63 (63.0%) | **0.680** |
| **Chronic SDH** | 0 (0.0%) | 2 (3.1%) | 2 (2.0%) | **0.295** |
| **Mixed Density SDH** | 0 (0.0%) | 7 (10.8%) | 7 (7.0%) | **0.044** |
| **Traumatic ICH[3]** | 9 (25.7%) | 11 (16.9%) | 20 (20.0%) | **0.295** |
| **Traumatic SAH[4]** | 12 (34.3%) | 21 (32.3%) | 33 (33.0%) | **0.841** |
| **Epidural Hemorrhage** | 1 (2.9%) | 0 (0.0%) | 1 (1.0%) | **0.171** |
| **GCS[5] at Presentation** |  |  |  | **0.520** |
| **14–15** | 32 (91.4%) | 57 (89.1%) | 89 (89.9%) |  |
| **9–13** | 1 (2.9%) | 5 (7.8%) | 6 (6.1%) |  |
| **3–8** | 2 (5.7%) | 2 (3.1%) | 4 (4.0%) |  |
| **Intubated on Arrival** | 2 (5.7%) | 2 (3.1%) | 4 (4.0%) | **0.521** |
| **Admission Destination** |  |  |  | **0.880** |
| **Floor** | 13 (81.2%) | 32 (82.1%) | 90 (77.6%) |  |
| **Intensive Care Unit** | **3 (18.8%)** | **6 (15.4%)** | **23 (19.8%)** |  |

[1]Computed Tomography

[2]Subdural Hematoma

[3]Intracerebral Hemorrhage

[4]Subarachnoid Hemorrhage

[5]Glasgow Coma Score

Data regarding discharge disposition are summarized in **Fig 3**. Forty-four patients (44%) had a unfavorable discharge disposition, and high frailty patients were significantly more likely to be discharged to an unfavorable location than low frailty patients (53.8% vs 25.7%, $p = 0.007$). Sex, antiplatelet/anticoagulant use, and presenting GCS category were not significantly associated with unfavorable discharge disposition. In multivariate analysis, increasing age (aOR 1.11, 95CI [1.04, 1.20], p = 0.004) and high frailty status (aOR 2.63, 95CI [1.02, 7.14], p = 0.005) were significantly associated with unfavorable discharge.

## Discussion

TBI is increasingly prevalent among the elderly. Accumulated evidence from both the trauma and orthopedic surgery literature suggests that age alone is an incomplete measure of vulnerability and decline after traumatic injury. The relationship between frailty metrics, which may be either physiologic or cumulative deficit models, and outcomes such as unfavorable discharge disposition, inpatient complications, and readmissions has been well established in these contexts. We hypothesized that physiologic frailty, as defined by the FRAIL score, would be associated with adverse short-term outcomes in the geriatric TBI population. In a cohort of patients older the age of 70 admitted to the neurosurgical service at a Level I trauma center, we find that physiologic frailty was associated with unfavorable discharge disposition independently of age on multivariate analysis. We did not, however, observe a relationship between

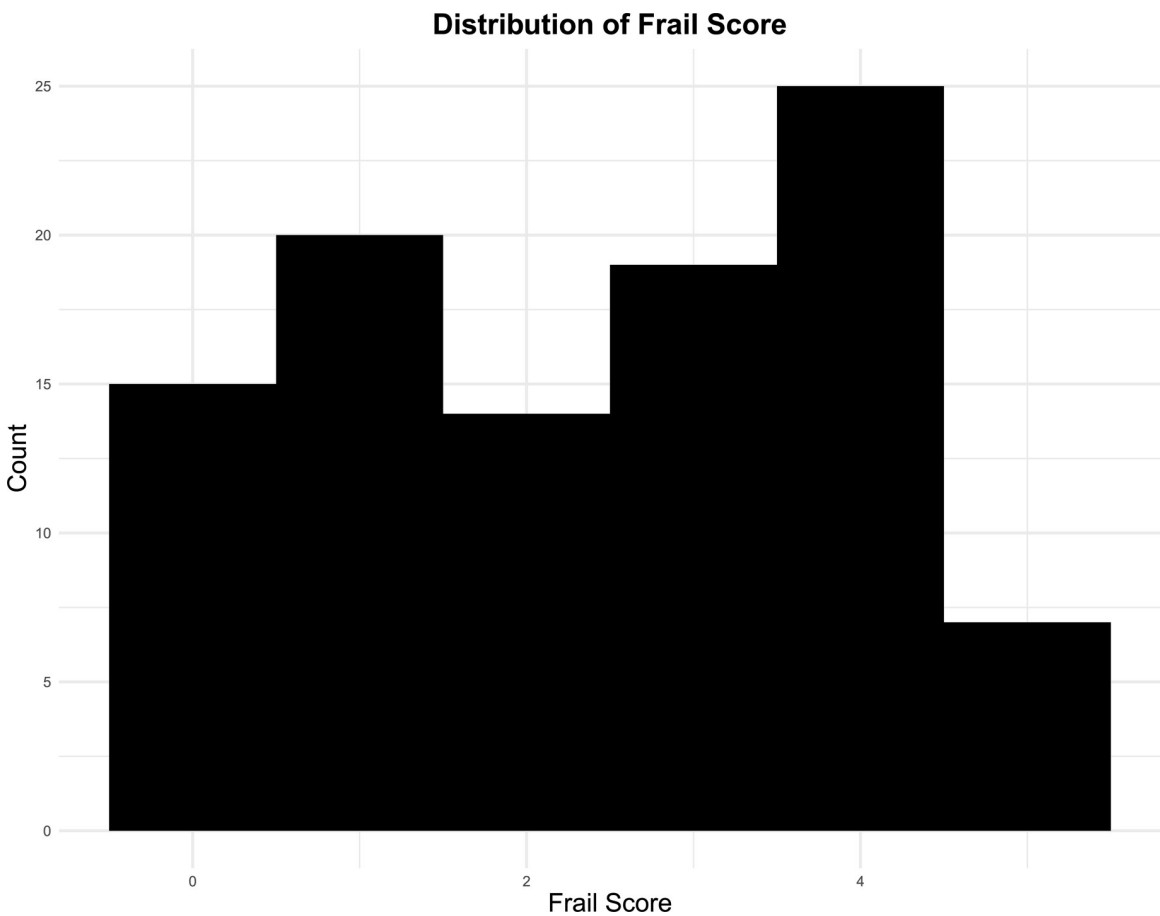

**Fig 2. Histogram demonstrating distribution of FRAIL scores in the patient cohort.**

physiologic frailty and major inpatient complications, consulting services per patient, LOS, discharge GCS, or readmission/re-presentation to acute care after discharge on univariate, uncontrolled analyses. Notably, in this cohort, the overall rate of inpatient complications and readmissions were low.

Our finding that frail patients are at increased odds of unfavorable discharge disposition contributes to a growing body of literature regarding frailty and trauma. For elderly patients, discharge to SNFs after acute hospitalization is itself associated with increased odds of 1-year mortality and post-discharge readmission [26]. Up to 40% of patients discharged to SNF never return home [26]. The strong association of frailty and unfavorable discharge disposition has been established in single-institutional cohorts of geriatric patients admitted to medical, trauma surgery, and orthopedic services, with or without surgical intervention [13, 24, 27–29]. In the context of our patient cohort, it may be the case that many community-dwelling frail adults would be triaged to a SNF even independently of an acute traumatic injury; nevertheless, it is also likely that mental and physiologic decompensation limits the ability of these patients to recover to functional baseline in both the short and long term [21, 30]. The high incidence of inpatient delirium among hospitalized frail elderly patients, which we were not able to quantify in our study, likely also contributes significantly to this disparity in discharge outcomes [15].

Our secondary findings that frail patients were not at risk of other adverse outcomes is unique in the geriatric trauma literature and merits further analysis. In comparison to the

**Table 2.  Inpatient and post-discharge outcomes for patient cohort.**

| | Low Frailty (N = 35) | High Frailty (N = 65) | Total (N = 100) | p value |
|---|---|---|---|---|
| **Major Inpatient Complication** | 1 (2.9%) | 6 (9.2%) | 7 (7.0%) | **0.233** |
| **Consulting Services per Patient** | 0.886 | 1.108 | 1.030 | **0.290** |
| **Patient Transferred to Another Service Prior to Discharge** | 0 (0.0%) | 7 (10.9%) | 7 (7.1%) | **0.042** |
| **Length of Stay [mean (range)]** | 3.46 (0,16) | 3.95 (1,19) | 3.78 (0,19) | **0.181** |
| **GCS at Discharge** | | | | **0.173** |
| **14–15** | 34 (100.0%) | 56 (90.3%) | 90 (93.8%) | |
| **9–13** | 0 (0.0%) | 2 (3.2%) | 2 (2.1%) | |
| **3–8** | 0 (0.0%) | 4 (6.5%) | 4 (4.2%) | |
| **Readmission within 30 days** | 3 (8.6%) | 9 (15.8%) | 12 (13.0%) | **0.318** |
| **Re-presentation to ED without hospital admission within 30 days** | 4 (11.4%) | 8 (13.6%) | 12 (12.8%) | **0.765** |
| **Discharge Disposition** | | | | |
| **Home** | 21 (60.0%) | 22 (33.8%) | 43 (43.0%) | |
| **Inpatient Rehabilitation** | 4 (11.4%) | 5 (7.7%) | 9 (9.0%) | |
| **Skilled Nursing Facility** | 9 (25.7%) | 29 (44.6%) | 38 (38.0%) | |
| **Long-Term Acute Care** | 0 (0.0%) | 0 (0.0%) | 0 (0.0%) | |
| **Home with Hospice** | 0 (0.0%) | 2 (3.1%) | 2 (2.0%) | |
| **Hospice** | 0 (0.0%) | 4 (6.2%) | 4 (4.0%) | |
| **Expired in Hospital** | 1 (2.9%) | 3 (4.6%) | 4 (4.0%) | |
| **Unfavorable Discharge** | 9 (25.7%) | 35 (53.8%) | 44 (44.0%) | **0.007** |
| **Neurosurgical Follow Up within 30 days** | **18 (51.4%)** | **27 (46.6%)** | **45 (48.4%)** | **0.648** |

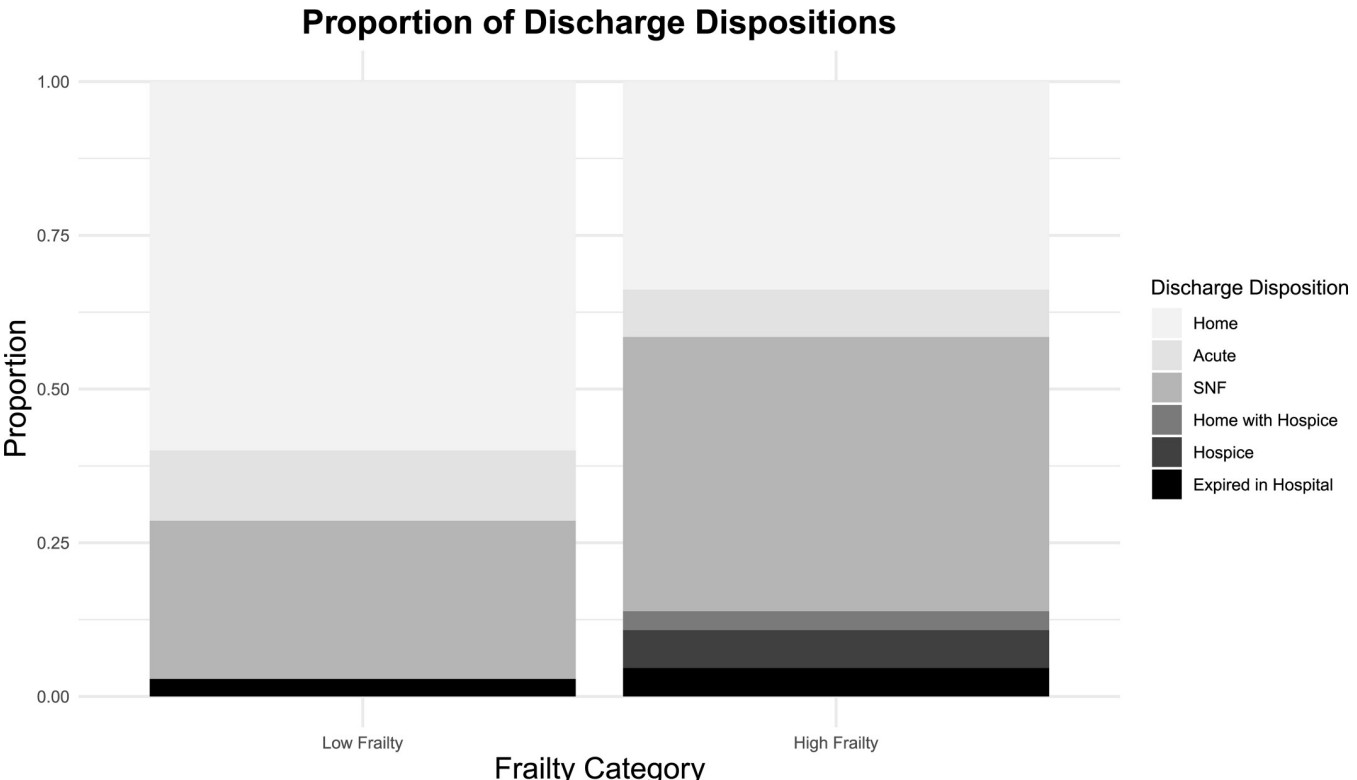

**Fig 3. Place of residence at admission and discharge outcomes stratified by frailty status.**

analysis of Gleason *et al*, who used the FRAIL score to assess outcomes in geriatric patients with operative orthopedic fractures, the distribution of frailty, in which a small minority of patients were categorized as non-frail, is largely comparable [24]. The resource intensity of inpatient care required for patients in this cohort differs significantly from those in trauma surgery or orthopedic cohorts. Mean LOS for geriatric TBI patients was approximately 4 days; in contrast, patients admitted with general or orthopedic trauma had mean LOS that ranges from 3–11 days [13, 17, 24, 28]. The overall rate of major inpatient complications, excluding delirium but including transfer to ICU, in our patients was 7%, which compares favorably to 28% in operative orthopedic trauma patients (though it should be noted this Fig includes delirium as a complication) and 20–30% in mixed trauma surgery patients (many of whom had non-operative injuries) [13, 15, 17, 24, 29]. It is worth noting that the true incidence of TBI in mixed trauma cohorts is not formally reported in many comparable studies; however, they are not explicitly excluded by any study and are as high as 30% in at least one study [28]. These findings call into question whether frailty, which is a measure of physiologic but not neurologic reserve, or major inpatient non-neurologic complications, is the most appropriate measures of inpatient morbidity for geriatric patients with TBI. Two contemporary assessments of frailty in the context of TBI noted dramatic differences between the frail and the nonfrail with regard to a variety of functional, psychosocial, and emotional outcomes [21, 22]; however, as compared to our analysis, these outcomes reflect relatively long-term sequelae of TBI as opposed to short-term or in-hospital complications, which are commonly the target of efficacious and cost-effectiveness frailty pathway interventions.

Regardless of frailty status, the low overall rate of follow-up with neurosurgical care after discharge remains significant. The rate of follow-up by 3 months (49%) in our patients, the majority of whom presented with mild TBI and all of whom were admitted to the hospital, actually compares well with follow-up data from the multicenter TRACK-TBI prospective trial, in which only 44% patients had seen any health care practitioner at 3 months [31]. Follow-up rates were similarly low at 9 of the 11 TRACK-TBI sites, except for 2 that had dedicated TBI clinics (at present, our hospital does not). Crucially, the likelihood of attending follow-up did not correlate insurance status, admission status (as compared to discharge from the emergency department), or presence of moderate-to-severe concussive symptoms at the 3-month mark. In context, the rate of follow up for geriatric TBI patients is actually lower than that of all discharged Medicare beneficiaries; with respect to a possible association between poor functional outcomes and frailty, one may suspect these patients may have unmet outpatient clinical needs [32]. Similarly, while the benefits of integrated geriatric care programs have the well-established effects of minimizing inpatient delirium, inpatient complications, and readmission rates, documented improvements in long-term functional outcomes may in fact be the most consequential basis upon which an inpatient frailty pathway for geriatric TBI patients could be established [15, 18].

## Limitations

This study has limitations. It is a retrospective series of patients admitted to a neurosurgical service of a single hospital; therefore, patients who re-present to or receive follow up in other health systems were outside the scope of our review. The patients are drawn from a single geographic region and therefore may not reflect the aggregate mix of patients, as determined by race, socioeconomic status, or other factors, seen in other centers around the country. Patients were included retrospectively and non-consecutively; however, it should be noted that even prospective assessments of frailty are not immune to selection bias given exclusion criteria (such as excluding patients who do not speak English or those who cannot answer questions

about their functional status at time of discharge). Additionally, patients in this particular study were excluded if they underwent operative intervention. While this decision was made to limit contributions to various in-hospital and post-discharge outcomes by the physiologic stress of surgery, it necessarily introduces selection bias by selecting either for patients with minor injuries or patients with possibly operative injuries in whom intervention was forgone on account of medical comorbidity or goals of care. As previously noted, the FRAIL score is just one of many measures of frailty but may not be the optimal tool to assess persistent and combined neurological and physiologic derangements that affect short and long-term functional status in this population.

## Conclusion

Frailty is independently associated with increased odds of unfavorable discharge disposition for geriatric patients admitted with TBI. The rate of follow-up, as seen in other studies, is low. Our study suggests that increased attention to the development of inpatient and outpatient care pathways and patient medical and social navigation protocols that more carefully assess alterations in neurological and physiologic derangement, and which identify and insure more complete and durable return to an optimal level of function is warranted.

## Acknowledgments

The authors would like to thank the following individuals, who assisted significantly with the clinical care of these patients:

Felicia Sun, MD MPhil
Hael Abdulrazeq, MD
Elias Shaaya, MD
Kerri Amaral, NP
Jenny Healey, PA-C
Brittany Hogan, PA-C
Monica Souza, PA-C
Emily Sullivan, NP
Jeremy Steinberg, PA-C
Allison Stocks, NP
James Taft, PA-C

## Author Contributions

**Conceptualization:** Rahul A. Sastry, Belinda Shao, Lynn McNicoll, Robert J. Weil, Ziya L. Gokaslan.

**Data curation:** Rahul A. Sastry, Josh R. Feler.

**Formal analysis:** Rahul A. Sastry, Josh R. Feler.

**Investigation:** Rahul A. Sastry, Josh R. Feler.

**Methodology:** Rahul A. Sastry.

**Project administration:** Rahul A. Sastry, Ziya L. Gokaslan.

**Resources:** Rahul A. Sastry.

**Software:** Rahul A. Sastry, Josh R. Feler.

**Supervision:** Rahul A. Sastry, Robert J. Weil.

**Validation:** Rahul A. Sastry, Josh R. Feler.

**Visualization:** Rahul A. Sastry, Josh R. Feler.

**Writing – original draft:** Rahul A. Sastry, Robert J. Weil.

**Writing – review & editing:** Rahul A. Sastry, Josh R. Feler, Belinda Shao, Rohaid Ali, Lynn McNicoll, Albert E. Telfeian, Adetokunbo A. Oyelese, Robert J. Weil, Ziya L. Gokaslan.

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
