## [Decision Letter · Decision Letter 0]

3 Aug 2022

PONE-D-22-19585Frailty Independently Predicts Unfavorable Discharge in Non-Operative Traumatic Brain Injury: A Retrospective Single-Institution Cohort StudyPLOS ONE

Dear Dr. Sastry,

Thank you for submitting your manuscript to PLOS ONE. After careful consideration, we feel that it has merit but does not fully meet PLOS ONE’s publication criteria as it currently stands. Therefore, we invite you to submit a revised version of the manuscript that addresses the points raised during the review process.

The reviewer #1 raises several essential concerns before the paper can be accepted for publication. Especially, I agree with the major comments 2. As the reviewer #1 recommended, I suggest the authors exclude patients who came from the “unfavorable” facilities because these patients usually cannot avoid “unfavorable discharge”.

We look forward to receiving your revised manuscript.

Kind regards,

Yusuke Tsutsumi

Academic Editor

PLOS ONE

Journal Requirements:

"NO

Reviewers' comments:

Reviewer's Responses to Questions

**Comments to the Author**

1. Is the manuscript technically sound, and do the data support the conclusions?

Reviewer #1: No

Reviewer #2: Yes

2. Has the statistical analysis been performed appropriately and rigorously? 

Reviewer #1: Yes

Reviewer #2: Yes

3. Have the authors made all data underlying the findings in their manuscript fully available?

Reviewer #1: No

Reviewer #2: Yes

4. Is the manuscript presented in an intelligible fashion and written in standard English?

Reviewer #1: Yes

Reviewer #2: Yes

5. Review Comments to the Author

Reviewer #1: The authors investigated an association of frailty in elderly patients suffered traumatic brain injury with disposition after hospital discharge. They found an association between frailty and the deterioration of outcomes. This research topic is valuable because frailty is one of the most important and urgent issues in the trauma and critical care today. However, several significant modifications are required for acceptance.

Major comments

1. The authors need to show the patient selection flow. Even if the study was conducted nonconsecutively, the total number of eligible patients needs to be shown first, then the number of patients who dropped out for each reason, and finally the number of patients included in the study. In addition, they should clearly indicate the number of patients excluded from the analysis due to in-hospital death.

2. I have concerns that the inclusion criteria may be inappropriate. To examine the association between frailty and unfavorable outcomes, they need to exclude patients admitted from facilities judged unfavorable. I think patients admitted from the Skilled Nursing Facility would not discharge home.

3. The authors treat the frail category as an ordinal variable. I am not sure if there is a difference between pre-frail and non-frail from this result. I perceive that the difference in the number of outcomes might just be based on differences in prehospital residence. Please clarify the characteristics of the distribution of those variables (and it would be better to exclude prehospital “unfavorable” facility). It might be better to treat frail as a binomial variable (non-frail or worse than prefrail) or as a categorical variable.

4. I recommend that variables for multivariate analysis be selected based on their clinical importance, not on their p-values. (Sun GW, Shook TL, Kay GL. Inappropriate use of bivariable analysis to screen risk factors for use in multivariable analysis. Journal of Clinical Epidemiology. 1996;49(8):907-916. doi:10.1016/0895-4356(96)00025-X)

5. In several places in the Abstract and Discussion, the authors describe primary outcomes, which are adjusted analyses, and secondary outcomes, which are unadjusted analyses, as being equivalent. I would suggest that the authors do not assert that the results of the unadjusted analysis are associated with the outcomes with the same strength as the results of the adjusted analysis.

Line 37: “Frailty was not associated with increased readmission rate, LOS, or rate of complications.”

Line 39: “Frailty is associated with increased odds of unfavorable discharge disposition but not with other major complications for geriatric patients admitted with TBI.”

Line 165: ” In a cohort of patients older the age of 70 admitted to the neurosurgical　service at a Level I trauma center, we find that physiologic frailty was associated with unfavorable　discharge disposition independently of age. We did not, however, observe a relationship between physiologic frailty and major inpatient complications, consulting services per patient, LOS, discharge GCS, or readmission/re-presentation to acute care after discharge. “

L183: “Our finding that frail patients are at risk for unfavorable discharge disposition but not other adverse outcomes”

L236: “Frailty is associated with increased odds of unfavorable discharge disposition but not with other major complications for geriatric patients admitted with TBI.”

6. Based on the journal's publication criteria, part of the conclusions would be inappropriate because it is unclear from which results they are derived.

Line 238: “Our study suggests that increased attention to the development of inpatient and outpatient care pathways and patient medical and social navigation protocols that more carefully assess alterations in neurological and physiologic derangement, and which identify and insure more complete and durable return to an optimal level of function is warranted.”

Minor comments

1. Line 21: The authors need to spell out “TBI”.

2. Line 22: It would be better to add elderly or geriatric patients as a target population.

3. Line 94: The authors need to define "number of consulting services provided to hospitalized patients".

4. Line 95: The citation for the definition of "major complications during hospitalization" needs to be mentioned.

5. The authors would do well to discuss the possibility that the indication for surgery causes selection bias (patients too severely ill to undergo surgery).

6. The authors need to describe that they treated GCS as categorical variables and the rationale of the category in Methods.

7. Please resubmit the Figure1 because of poor resolution.

Reviewer #2: It is not surprising that the percentage of FRAIL increases with age. And it is also natural that the frail will have a nonfavorable discharge disposition. However, it is also important to show the obvious, as it will lead to the next study.

As the author writes in LIMITATION, this study has very little external validity.

6. PLOS authors have the option to publish the peer review history of their article (what does this mean?). If published, this will include your full peer review and any attached files.

Reviewer #1: No

Reviewer #2: **Yes: **Fumihito Ito

---

## [Author Response · Author response to Decision Letter 0]

30 Aug 2022

Dear Dr. Tsutsumi, 

Thank you for considering our manuscript and for providing us with this specific feedback. We have made the following clarifications and/or improvements to the manuscript in order to appropriately address all the points that were raised. Specifically, we have attempted to ensure that the manuscript aligns with PLOS formatting requirements and that the statement regarding informed consent is appropriate. The authors received no specific funding for this work.

We would like to note that, in response to comment #2, we have refined the inclusion criteria of this study to include only patients who were admitted from home (and thus excluding the small number of patients who came from locations that would have been classified as unfavorable discharge destinations). The authors would like to note that, at least for an American audience, discharge metrics such as percentage of patients discharged to nursing facilities are likely relevant regardless of what locations the patients arrive from. Nevertheless, the clinically relevant result of this study is robust to this change in cohort definition. 

The authors investigated an association of frailty in elderly patients suffered traumatic brain injury with disposition after hospital discharge. They found an association between frailty and the deterioration of outcomes. This research topic is valuable because frailty is one of the most important and urgent issues in the trauma and critical care today. However, several significant modifications are required for acceptance.

Major comments

1. The authors need to show the patient selection flow. Even if the study was conducted nonconsecutively, the total number of eligible patients needs to be shown first, then the number of patients who dropped out for each reason, and finally the number of patients included in the study. In addition, they should clearly indicate the number of patients excluded from the analysis due to in-hospital death.

 A new figure 1, which demonstrates this patient selection flow, has been included. 

2. I have concerns that the inclusion criteria may be inappropriate. To examine the association between frailty and unfavorable outcomes, they need to exclude patients admitted from facilities judged unfavorable. I think patients admitted from the Skilled Nursing Facility would not discharge home.

As previously noted, the authors re-did the relevant analyses excluding patients who were admitted from unfavorable locations. While the fundamental conclusion of the analysis is unchanged, this change does eliminate a major potential source of bias in the results. As such, the authors have implemented this change as part of the cohort definition rather than a subgroup analysis. 

3. The authors treat the frail category as an ordinal variable. I am not sure if there is a difference between pre-frail and non-frail from this result. I perceive that the difference in the number of outcomes might just be based on differences in prehospital residence. Please clarify the characteristics of the distribution of those variables (and it would be better to exclude prehospital “unfavorable” facility). It might be better to treat frail as a binomial variable (non-frail or worse than prefrail) or as a categorical variable.

The authors have also implemented this change. Based on the distribution of FRAIL scores, we chose to dichotomize the population into low FRAIL (0-1) and high FRAIL (2-5) categories. As previously noted, patients with prehospital “unfavorable” status have been excluded. 

4. I recommend that variables for multivariate analysis be selected based on their clinical importance, not on their p-values. (Sun GW, Shook TL, Kay GL. Inappropriate use of bivariable analysis to screen risk factors for use in multivariable analysis. Journal of Clinical Epidemiology. 1996;49(8):907-916. doi:10.1016/0895-4356(96)00025-X)

The authors broadly agree with this statement, and have updated the chosen multivariate model to reflect this change. 

5. In several places in the Abstract and Discussion, the authors describe primary outcomes, which are adjusted analyses, and secondary outcomes, which are unadjusted analyses, as being equivalent. I would suggest that the authors do not assert that the results of the unadjusted analysis are associated with the outcomes with the same strength as the results of the adjusted analysis.

Line 37: “Frailty was not associated with increased readmission rate, LOS, or rate of complications.” 

Line 39: “Frailty is associated with increased odds of unfavorable discharge disposition but not with other major complications for geriatric patients admitted with TBI.” 

Line 165: ” In a cohort of patients older the age of 70 admitted to the neurosurgical　service at a Level I trauma center, we find that physiologic frailty was associated with unfavorable　discharge disposition independently of age. We did not, however, observe a relationship between physiologic frailty and major inpatient complications, consulting services per patient, LOS, discharge GCS, or readmission/re-presentation to acute care after discharge. “

L183: “Our finding that frail patients are at risk for unfavorable discharge disposition but not other adverse outcomes”

L236: “Frailty is associated with increased odds of unfavorable discharge disposition but not with other major complications for geriatric patients admitted with TBI.”

The language of these statements have been modified to more clearly distinguish between the primary result, which is controlled for on multivariate analysis, and the secondary results, which are not. 

6. Based on the journal's publication criteria, part of the conclusions would be inappropriate because it is unclear from which results they are derived. 

Line 238: “Our study suggests that increased attention to the development of inpatient and outpatient care pathways and patient medical and social navigation protocols that more carefully assess alterations in neurological and physiologic derangement, and which identify and insure more complete and durable return to an optimal level of function is warranted.”

The conclusion has also been amended, as above. 

Minor comments

1. Line 21: The authors need to spell out “TBI”. 

This abbreviation has now been spelled out.

2. Line 22: It would be better to add elderly or geriatric patients as a target population.

This clarification has been added 

3. Line 94: The authors need to define "number of consulting services provided to hospitalized patients".

A parenthetical listing some possible consulting services has been added to clarify this statement.

4. Line 95: The citation for the definition of "major complications during hospitalization" needs to be mentioned.

An appropriate citation has been included. 

5. The authors would do well to discuss the possibility that the indication for surgery causes selection bias (patients too severely ill to undergo surgery).

Additional discussion of this point has been added to the limitations section of the discussion

6. The authors need to describe that they treated GCS as categorical variables and the rationale of the category in Methods.

This has been clarified in the manuscript

7. Please resubmit the Figure1 because of poor resolution.

The old figure 1 (now figure 3) has been updated. 

The authors would again like to thank the editorial staff for giving them the chance to revise this manuscript. 

Sincerely,

Rahul A Sastry

On behalf of all other authors

---

## [Decision Letter · Decision Letter 1]

21 Sep 2022

Frailty Independently Predicts Unfavorable Discharge in Non-Operative Traumatic Brain Injury: A Retrospective Single-Institution Cohort Study

PONE-D-22-19585R1

Dear Dr. Sastry,

We’re pleased to inform you that your manuscript has been judged scientifically suitable for publication and will be formally accepted for publication once it meets all outstanding technical requirements.

Kind regards,

Yusuke Tsutsumi

Academic Editor

PLOS ONE

Additional Editor Comments (optional):

Reviewers' comments:

Reviewer's Responses to Questions

**Comments to the Author**

1. If the authors have adequately addressed your comments raised in a previous round of review and you feel that this manuscript is now acceptable for publication, you may indicate that here to bypass the “Comments to the Author” section, enter your conflict of interest statement in the “Confidential to Editor” section, and submit your "Accept" recommendation.

Reviewer #1: All comments have been addressed

Reviewer #2: All comments have been addressed

2. Is the manuscript technically sound, and do the data support the conclusions?

Reviewer #1: Yes

Reviewer #2: Partly

3. Has the statistical analysis been performed appropriately and rigorously? 

Reviewer #1: Yes

Reviewer #2: Yes

4. Have the authors made all data underlying the findings in their manuscript fully available?

Reviewer #1: No

Reviewer #2: Yes

5. Is the manuscript presented in an intelligible fashion and written in standard English?

Reviewer #1: Yes

Reviewer #2: Yes

6. Review Comments to the Author

Reviewer #1: The authors have responded appropriately to the remarks made by the reviewer. No further comments are noted.

Reviewer #2: (No Response)

7. PLOS authors have the option to publish the peer review history of their article (what does this mean?). If published, this will include your full peer review and any attached files.

Reviewer #1: No

Reviewer #2: No

---

## [Editor Report · Acceptance letter]

27 Sep 2022

PONE-D-22-19585R1 

Frailty Independently Predicts Unfavorable Discharge in Non-Operative Traumatic Brain Injury: A Retrospective Single-Institution Cohort Study 

Dear Dr. Sastry:

I'm pleased to inform you that your manuscript has been deemed suitable for publication in PLOS ONE. Congratulations! Your manuscript is now with our production department. 

Kind regards, 

on behalf of

Dr. Yusuke Tsutsumi 

Academic Editor

PLOS ONE